# Spatial variations and predictors of neonatal mortality among births to HIV-infected and non-infected mothers in rural Zambia

Samson Shumba[1]*, Deborah Tembo[2], Miyanda Simwaka[3], Nedah Chikonde Musonda[4], Chipo Nkwemu[1,5], Sebean Mayimbo[6]

1 Department of Epidemiology and Biostatistics, School of Public Health, University of Zambia, Lusaka, Zambia, 2 Department of Emergency Preparedness and Response, Zambia National Public Health Institute, Lusaka, Zambia, 3 Ministry of Health, Lusaka, Zambia, 4 Copper Rose Zambia, Lusaka, Zambia, 5 Centre for Infectious Disease Research in Zambia, Lusaka, Zambia, 6 Department of Midwifery, Women's and Child Health, School of Nursing Sciences, University of Zambia, Lusaka, Zambia

* Samsonshumba1@gmail.com

## Abstract

In Zambia, neonatal mortality still remains a noteworthy public health problem with a current rate of 27 deaths per 1000 and ranking 162 out of 195 countries globally. The study aimed to investigate the spatial variations and predictors of neonatal mortality in rural Zambia among HIV-infected and non-infected mothers using the national-level data from the 2018 Zambia Demographic and Health Survey (ZDHS). Statistical analyses were conducted using the Rao – Scott Chi-square test to assess associations between neonatal mortality and categorical variables. Additionally, a multilevel mixed effect logistic regression model was used to examine predictors of neonatal mortality. Geospatial variations of neonatal mortality across Zambia's ten provinces were mapped using Quantum Geographical Information System (QGIS) version 3.34.1. Data analysis was performed using R and Stata version 14.2. This study examined the spatial variations and predictors of neonatal mortality among HIV-infected and non-infected mothers in rural Zambia using the 2018 Zambia Demographic Health Survey dataset. Key findings include the protective role of maternal education, with those having secondary or higher education showing reduced odds of neonatal mortality. Women aged 20–24 years in the study had higher odds of neonatal death compared to younger mothers, while delivering in public health facilities was associated with increased neonatal mortality. Maternal HIV status had no significant impact on neonatal outcomes. Spatial analysis revealed significant regional disparities, with high mortality rates in Central, Southern, and Eastern provinces, while North Western Province had lower rates. These results emphasize the need for improved healthcare quality, targeted maternal education programs, and region-specific interventions to address neonatal mortality in Zambia.

**Data availability statement:** Authorization to utilize the data was secured from ICF Macro, accessible at https://dhsprogram.com/data, under the dataset titled ZMKR71FL.DTA and ZMAR71FL.DTA

**Funding:** The authors received no specific funding for this work.

**Competing interests:** The authors have declared that no competing interests exists.

## Introduction

Neonatal mortality is characterized by the death of a live-born infant, irrespective of the gestational age at birth, within the initial 28 completed days of life [1]. The first month following birth is a critical period for newborn survival. In 2020, this phase saw the tragic loss of 2.4 million infants worldwide [2]. During that same year, nearly 47% of all deaths in children under the age of five occurred during the neonatal period, which spans the first 28 days of life and 98% of the deaths occurred in low and middle-income countries (LMIC) [1–3]. This proportion reflects a notable increase from the 40% reported in 1990, according to WHO (2024) [1]. The world has made impressive progress towards the survival of children since 1990. In 2020, the number of neonatal deaths declined from 5 million in 1990 to 2.4 million, nonetheless, the decline in neonatal mortality from 1990 to 2020 has been slower than that of the post-neonatal under-five mortality [1].

Sub-Saharan Africa bears the highest global neonatal mortality rate, reaching approximately 27 deaths per 1000 live births, and plays a significant role in global newborn fatalities, accounting for 43% of the total [4,5]. Central and southern Asia report a neonatal mortality rate of 23 deaths per 1000 live births, contributing to 36% of global newborn deaths [1]. Moreover, a child born in sub-Saharan Africa faces a staggering tenfold higher likelihood of dying in the first month than a child born in a high-income country. In Uganda a study revealed that the underlying causes of death are related to poor access and low utilization of health services during pregnancy and childbirth [6]. Furthermore, findings have revealed that more newborn deaths occur at home among the rural poor [7].

Zambia relative to other countries fairs poorly ranking 162 out of 195 countries globally in terms of neonatal mortality [8]. According to the Zambia Statistical Agency (2018), it indicated that neonatal mortality rate decreased from 37 to 27 neonatal deaths per 1000 live births between 2001 and 2018 [9]. In addition, numerous studies have highlighted a range of factors associated with increased neonatal mortality rates, both in the general population and within facility-based settings. These include lower levels of maternal education [10–12], place of delivery and residence [13,14], maternal age [15,16], parity [17], gestational age [10,15], inadequate antenatal visits [18,19], newborn sex [13], newborn age [19], low birth weight [20], maternal and fetal complications [20], preterm birth [21], maternal BMI [22] and hypothermia [23]. Despite these findings, there remains a gap in understanding neonatal mortality and its associated risk factors. Specifically, while studies have explored neonatal mortality among children born to HIV-infected mothers, particularly in urban areas [24], it is unclear whether HIV status acts as a covariate or modifier of mortality risk. This gap is especially evident in rural communities, where additional challenges may exacerbate the issue, such as disparities between HIV-infected and non-infected mothers in rural Zambia [25,26]. Therefore, this study examined the spatial variations and predictors of neonatal mortality among births to HIV-infected and non-infected mothers in rural Zambia.

## Conceptual framework

We modified the conceptual framework of neonatal mortality using the analytical framework established by Mosley and Chen. Originally proposed in 1984, this analytical framework for child mortality was subsequently adapted to emphasize variations in infant and child mortality attributable to socio-economic differentials, bio-demographic factors, household environmental conditions, nutrient deficiencies, and community influences (Fig 1) [27].

## Methods

This study constitutes a secondary analysis of microdata utilizing national-level data sourced from the Zambia Demographic and Health Survey (ZDHS) program using the survey dataset in 2018. The ZDHS is a comprehensive, nationally representative household survey conducted by the Zambia Statistics Agency in collaboration with global partners, including ICF International and the United States Agency for International Development (USAID). The survey employs a two-stage sampling process, initially selecting enumeration areas (EAs) and subsequently households.

## Dependent and independent variables

The primary outcome of interest in this study is neonatal mortality, defined as the death of infants within the first 28 days of life. The explanatory variables used are grouped into demographic, socio-economic, behavioral, and community-level factors to provide a holistic view of the determinants of neonatal mortality. Demographic variables include the mother's age (categorized as 15–19, 20–24, 25–29, 30–34, 35–39, 40–44, or 45–49 years), child's sex (male or female), and the number of children ever born (birth order). These variables capture biological and family composition factors that influence neonatal outcomes. Socio-economic factors encompass maternal education (none, primary, secondary/tertiary), household wealth index (poorest, poorer, middle, richer, richest), marital status (married or not married), and employment status (employed or unemployed). These indicators reflect the mother's access to resources and social determinants of health. Behavioral factors focus on health-related practices, including the duration of breastfeeding (ever vs. never breastfed). These behaviors are critical in understanding neonatal care practices that affect survival during the first month

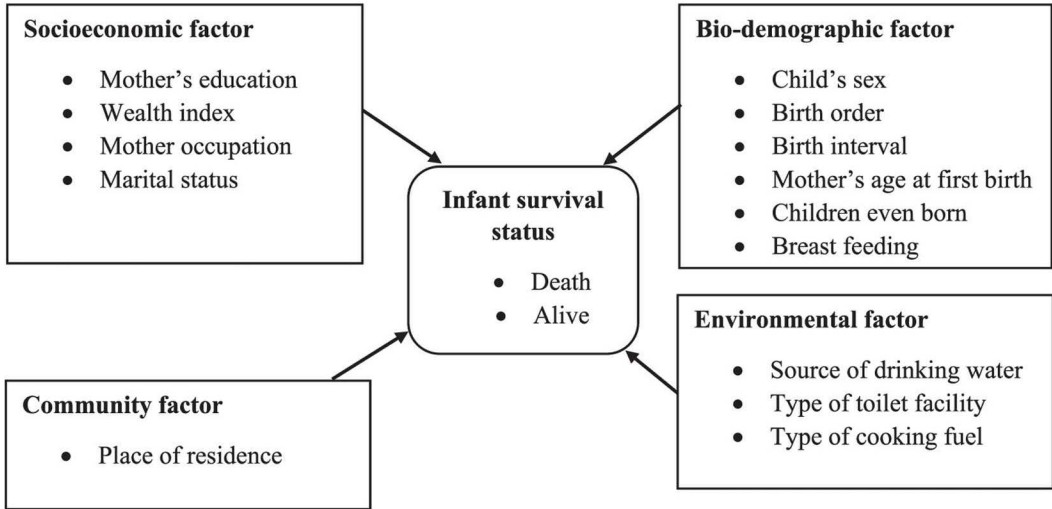

**Conceptual framework**

**Fig 1. Conceptual framework. Neonatal mortality conceptual framework.** (Source: Mosley and Chen, 2003).

of life. Community-level factors consist of the source of drinking water (improved or unimproved), place of delivery (home, private, or public health facility) and region. These variables capture the environmental and infrastructural context in which the mother and newborn reside, which can significantly affect health outcomes [28].

The selection of these variables demographic, socio-economic, environmental, and community-level factors aligns with existing research on neonatal mortality and reflects key determinants recognized by global health organizations such as the World Health Organization (WHO) and UNICEF. Each variable captures essential dimensions that influence maternal and child health outcomes, which are critical to informing policies and interventions [29–31].

## Data analysis

For descriptive analysis, frequencies and percentages were calculated for categorical variables to summarize the data. To assess the association between neonatal mortality (the outcome variable) and categorical predictors, the Rao–Scott Chi-square test was employed, accounting for the complex survey design. A multilevel mixed-effects logistic regression model was used to identify individual and community-level factors influencing neonatal mortality, recognizing the hierarchical structure of the data. The random factor in the study was cluster number coded as "v001".

The study followed an investigator-led approach for variable selection, drawing on a comprehensive review of relevant literature to ensure the inclusion of theoretically and empirically relevant predictors. Model selection was guided by the Akaike Information Criterion (AIC) and Bayesian Information Criterion (BIC), with the Intra-Class Correlation (ICC) used to quantify the extent of variance explained at the community level. These measures ensured the identification of the most appropriate and parsimonious model. All analyses were performed using Stata version 14.2.

Proportions of neonatal mortality by maternal HIV status were generated in Stata using survey weights. These proportions were then mapped using Quantum Geographic Information System (QGIS) version 3.34.1, applying the World Geodetic System (WGS) 1984 Universal Transverse Mercator (UTM) Zone 36S coordinate reference system to ensure accurate geographic projection. For spatial autocorrelation analysis, the proportions were imported into R version 4.3.2 and analyzed at the DHS cluster level using Moran's I statistic. The analysis in R utilized the *spdep*, *sf*, *rgdal*, and *tmap* packages, which facilitated the construction of spatial weights, management of geospatial data, and visualization of spatial patterns of neonatal mortality by maternal HIV status.

## Model specification for multilevel mixed effect logistic regression

$$logit\left(P\left(Y_{ij}=1\right)\right) = \beta_0 + \sum_{k=1}^{k} \beta_k X_{ijk} + u_j$$

Where:

Logit is the log-odds function:

$$logit\left(P\left(Y_{ij}=1\right)\right) = \ln\left(\frac{P\left(Y_{ij}=1\right)}{1-P\left(Y_{ij}=1\right)}\right)$$

$u_j \sim N\left(0, \sigma_u^2\right)$ assumes that the cluster-level random effect follows a normal distribution with mean 0 and variance $\sigma_u^2$

Let:

- $Y_{ij}$= Outcome for individual $i$ in cluster $j$ (binary: $Y_{ij}=1$ if the event occurs, and $Y_{ij}=0$ otherwise).

- $X_{ij} = X_{ij1}, X_{ij2}, \ldots, X_{ijk}$ =vector of individual-level covariates

- $\beta_0 =$ Fixed intercept (baseline log-odds of the outcome).

- $\beta_k =$ Coefficients for individual-level predictors $X_{ijk}$

- $u_j =$ Random intercept for cluster jjj, accounting for unobserved heterogeneity among clusters [32].

**Intra-Class Correlation (ICC) in multilevel logistic regression**

The ICC quantifies the proportion of the total variance in the outcome attributable to differences between clusters:

$$ICC = \frac{\sigma_u^2}{\sigma_u^2 + \frac{\pi^2}{3}}$$

Here $\frac{\pi^2}{3}$ is the variance of the logistic distribution at the individual level [23,33].

**Spatial autocorrelation assessment using Moran's I**

$$I = \frac{N}{W} \cdot \frac{\sum_{i=1}^{N} \sum_{j=1}^{N} w_{ij}(x_i - \bar{x})}{\sum_{i=1}^{N}(x_i - \bar{x})}$$

*Where* :
$N = Number\ of\ provinces$
$x_i = Value\ of\ the\ variable\ at\ location\ i$
$\bar{x} = Mean\ of\ the\ x$
$w_{ij} = Spatial\ weight\ between\ units\ i\ and\ j\ (defines\ neighbor\ relations)$
$\sum_{i=1}^{N} \sum_{j=1}^{N} w_{ij} = Sum\ of\ all\ spatial\ weights$

**Ethics statement**

The methodologies used in the 2018 ZDHS, including biomarker measurement protocols, were ethically cleared by both the Inner City Fund (ICF) institutional review boards (IRBs) and the Tropical Diseases Research Centre (TDRC) in Zambia. Oral consent was obtained from respondents, with parental or guardian consent for adolescents under 18. Neonate information was collected from caregivers. Detailed information on the DHS consent process is available (https://www.dhsprogram.com/What-We-Do/Protecting-the-Privacy-of-DHS-Survey-Respondents.cfm). Authorization to use ZDHS data was granted by ICF Macro, and the dataset can be accessed (https://www.dhsprogram.com/data). The user strictly adhered to confidentiality guidelines, ensuring the anonymity of respondents.

## Results

Participation in the survey was limited to neonates aged one month from selected households of women who had consented to take part in the research. A total of 2015 participants were enrolled into the study comprising neonates from HIV-infected and non-infected mothers (Fig 2).

The map was generated using PowerPoint 2016. (Source: author generated). Dataset source: ZDHS women dataset accessible at: (https://www.dhsprogram.com/data).

The data presented in Table 1 provides an overview of neonatal mortality in rural settings among mothers infected with HIV and those not infected with HIV, drawing from the 2018 Zambia Demographic and Health Survey dataset.

The findings reveal distinct patterns in neonatal mortality across various background characteristics, including maternal age, education level, employment status, neonate gender, birth order, birthweight, place of delivery, and HIV status. Notably, neonatal mortality increases significantly with maternal age ($p < 0.0001$), with the highest mortality observed in the age group 45–49 (100%) among HIV-infected mothers compared to 95.21% in non-HIV infected mothers.

Furthermore, mortality rates were higher among HIV-infected mothers (54.56%) compared to non-HIV-infected mothers without formal education (40.78%). Overall, neonatal mortality stood at 41.13% among mothers (both HIV and non-HIV-infected) with no formal education. Neonatal mortality was significantly higher among HIV-infected mothers who were not working (49.39%), in contrast to non-HIV-infected mothers, where high mortality was observed among those who were employed (36.36%).The neonatal mortality rate was high at 37.17% among neonates whose mothers (both HIV infected and non-infected mothers) were reported to be working compared to those who were not working (26.14%). Neonatal mortality was also higher among males in both HIV-infected and non-infected mothers (52.99% and 33.24%, respectively). Moreover, neonatal mortality was consistently higher among neonates with a birth order of 7 or higher among HIV-infected and non-infected mothers in rural Zambia (81.15% and 61.82%), with overall mortality also reported high among those of the 7th or higher birth order (63.73%).

Additionally, neonatal mortality was reported higher among neonates born with a weight of 4000 grams or higher among HIV-infected and non-infected mothers (81.15% and 21.00%, respectively). Overall, neonatal mortality was higher among mothers infected with HIV (51.48%) compared to those not infected (29.89%).

## Spatial analysis of neonatal mortality among births to HIV-infected mothers in rural Zambia

The study examined the geospatial patterns of neonatal mortality among births to HIV-infected mothers in rural Zambia. It revealed that North-western and Eastern provinces exhibited a high proportion of neonatal mortality, followed by Western

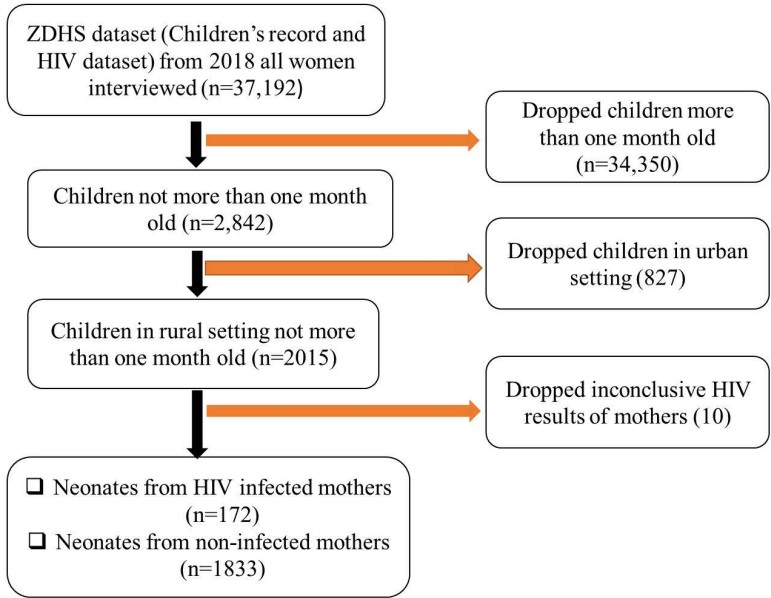

**Fig 2. Description of sample derivation criteria.**

**Table 1. Key attributes of neonatal mortality in rural setting of Zambia in 2018.**

| Variables | HIV – Infected (n = 172) | | Non-HIV infected (n = 1,833) | | Overall (n = 2,005) | |
|---|---|---|---|---|---|---|
| | **Alive** | **Dead** | **Alive** | **Dead** | **Alive** | **Dead** |
| **Age of mother** | <0.0001* | | 0.0001* | | <0.0001* | |
| 15 - 19 | 2 (100) | 0 (0.00) | 247 (94.28) | 15 (5.72) | 249 (94.54) | 15 (94.54) |
| 20 - 24 | 23 (70.71) | 9 (29.29) | 341 (83.58) | 67 (16.42) | 372 (82.73) | 78 (17.27) |
| 25 – 29 | 11 (59.26) | 8 (40.74) | 230 (77.87) | 65 (22.13) | 248 (76.61) | 76 (23.39) |
| 30 – 34 | 22 (70.25) | 9 (29.75) | 182 (69.47) | 79 (45.30) | 214 (70.11) | 91 (29.89) |
| 35 – 39 | 18 (59.28) | 12 (40.72) | 117 (54.70) | 97 (45.30) | 136 (54.73) | 113 (45.27) |
| 40 – 44 | 8 (26.76) | 21 (73.24) | 60 (38.01) | 98 (61.99) | 68 (35.63) | 122 (64.37) |
| 45 - 49 | 0 (0.00) | 18 (100.00) | 4 (4.79) | 81 (95.21) | 4 (3.94) | 99 (96.06) |
| **Education level of the mother** | 0.584 | | <0.0001* | | <0.0001* | |
| No Education | 12 (45.44) | 14 (54.56) | 127 (59.22) | 88 (40.78) | 148 (58.87) | 103 (41.13) |
| Primary | 44 (48.62) | 46 (51.38) | 723 (68.56) | 332 (31.44) | 786 (67.10) | 386 (32.90) |
| Secondary | 26 (61.51) | 16 (38.49) | 317 (79.79) | 80 (20.21) | 351 (77.80) | 100 (22.20) |
| Higher | 1 (50.64) | 1 (49.36) | 13 (77.54) | 4 (22.46) | 15 (75.30) | 5 (24.70) |
| **Marital status** | 0.6842 | | 0.1775 | | 0.3384 | |
| Not married | 32 (54.13) | 27 (45.87) | 279 (73.61) | 100 (26.39) | 319 (70.94) | 131 (29.06) |
| Married | 51 (49.94) | 50 (50.07) | 902 (69.09) | 404 (30.91) | 981 (67.93) | 463 (32.07) |
| **Wealth Index of the mother** | 0.92 | | 0.0821 | | 0.1169 | |
| Poorest | 25 (51.17) | 24 (48.83) | 456 (68.68) | 208 (31.32) | 501 (67.91) | 237 (32.09) |
| Poorer | 27 (56.59) | 21 (43.41) | 407 (75.12) | 135 (24.88) | 440 (73.27) | 160 (26.73) |
| Middle | 21 (50.44) | 20 (49.56) | 224 (64.91) | 121 (35.09) | 256 (63.89) | 145 (36.11) |
| Richer | 7 (44.18) | 8 (55.82) | 66 (65.34) | 35 (34.66) | 73 (62.62) | 43 (37.38) |
| Richest | 3 (41.28) | 5 (58.72) | 27 (86.17) | 4 (13.83) | 32 (77.92) | 9 (22.08) |
| **Mother currently working** | 0.8562 | | 0.0001* | | 0.0002* | |
| No | 41 (50.61) | 40 (49.39) | 678 (75.71) | 218 (24.29) | 739 (73.86) | 261 (26.14) |
| Yes | 41 (52.37) | 38 (47.63) | 503 (63.74) | 286 (36.26) | 562 (62.83) | 333 (37.17) |
| **Sex of child** | 0.1504 | | 0.0048* | | 0.0003* | |
| Male | 45 (47.01) | 51 (52.99) | 599 (66.76) | 298 (33.24) | 658 (64.77) | 358 (35.23) |
| Female | 37 (58.24) | 27 (41.76) | 582 (73.93) | 205 (26.07) | 642 (73.13) | 236 (26.87) |
| **Birth Order** | 0.0004* | | <0.0001 | | <0.0001* | |
| 1-3rd birth order | 37 (63.39) | 21 (36.61) | 682 (86.41) | 107 (13.59) | 741 (85.02) | 130 (14.98) |
| 4-6th birth order | 39 (61.11) | 25 (38.89) | 338 (71.39) | 136 (28.61) | 390 (70.17) | 166 (29.83) |
| 7th + order | 7 (18.85) | 32 (81.15) | 161 (38.18) | 261 (61.82) | 169 (36.27) | 298 (63.73) |
| **Place of Delivery** | 0.3923 | | 0.015* | | 0.0187* | |
| Home | 7 (81.33) | 1 (18.67) | 164 (83.53) | 32 (16.47) | 179 (83.87) | 34 (16.13) |
| Private health facility | 8 (100.00) | 0 (0.00) | 84 (95.45) | 4 (4.55) | 1003 (89.13) | 122 (10.87) |
| Government | 65 (79.26) | 17 (20.74) | 912 (89.91) | 102 (10.09) | 92 (95.83) | 4 (4.17) |
| Unspecified | 4 (100.00) | 0 (0.00) | 21 (82.61) | 4 (17.39) | 27 (81.92) | 36 (18.08) |
| **Birth weight** | 0.0004* | | <0.0001* | | <0.0001* | |
| Less than 2500 | 37 (63.39) | 21 (36.61) | 72 (80.83) | 17 (19.17) | 82 (80.71) | 20 (19.29) |
| 2500 - 3999 | 39 (61.11) | 25 (38.89) | 789 95.05) | 41 (4.95) | 862 (94.27) | 52 (5.73) |
| 4000+ | 7 (18.85) | 32 (81.15) | 319 (79.00) | 85 (21.00) | 357 (79.02) | 95 (20.98) |
| **ANC visits** | 0.7065 | | 0.6983 | | 0.8741 | |
| <8 ANC visits | 81 (86.97) | 12 (13.02) | 1142 (95.19) | 58 (4.81) | 1257 (94.59) | 72 (5.41) |
| 8+ANC visits | 1 (100.00) | 0 (0.00) | 17 (92.95) | 1 (7.05) | 19 (93.68) | 1 (6.32) |

*(Continued)*

**Table 1.** (Continued)

| Variables | HIV – Infected (n = 172) | | Non-HIV infected (n = 1,833) | | Overall (n = 2,005) | |
|---|---|---|---|---|---|---|
| | Alive | Dead | Alive | Dead | Alive | Dead |
| **HIV Results** | | | | | <0.0001* | |
| HIV negative | NA | NA | NA | NA | 1180.75 | 503 (29.89) |
| HIV positive | NA | NA | NA | NA | 83 (51.48) | 78 (48.52) |
| **Duration of breastfeeding** | <0.0001 | | <0.0001 | | <0.0001 | |
| Ever breastfed | 4 (25.03) | 13 (74.97) | 45 (38.37) | 72 (61.63) | 49 (36.66) | 85 (63.34) |
| Never breastfed | 1 (9.32) | 6 (90.68) | 5 (6.82) | 71 (93.18) | 6 (7.01) | 77 (92.99) |
| Still breastfeeding | 78 (100.00) | 0 (0.00) | 1131 (100.00) | 0 (0.00) | 1209 (100.00) | 0 (0.00) |
| **Children ever born** | 0.008 | | <0.0001 | | <0.0001 | |
| Zero to two | 24 (68.46) | 11 (31.54) | 530 (89.77) | 60 (10.23) | 71 (11.41) | 554 (88.59) |
| three to four | 26 (57.09) | 20 (42.91) | 279 (77.62) | 80 (22.38) | 100 (24.71) | 305 (75.29) |
| five to six | 25 (62.64) | 15 (37.36) | 211 (67.38) | 102 (32.62) | 117 (33.16) | 236 (66.84) |
| seven plus | 7 (18.85) | 32 (81.15) | 161 (38.18) | 261 (61.82) | 293 (63.49) | 168 (36.51) |
| **Source of drinking water** | 0.4797 | | 0.4745 | | 0.696 | |
| Improved | 61 (53.13) | 54 (46.87) | 625 (71.06) | 255 (29.94) | 309 (31.02) | 686 (68.98) |
| Unimproved | 21 (47.23) | 24 (52.77) | 556 (69.07) | 249 (30.93 | 273 (32.09) | 577 (67.91) |
| **Type of toilet facility** | 0.0727 | | 0.6089 | | 0.3667 | |
| Improved | 40(60.79) | 25 (39.21) | 428 (71.01) | 175 (28.99) | 467 (70.02) | 200 (29.98) |
| Unimproved | 43 (45.22) | 53 (54.78) | 753 (69.61) | 329 (30.39) | 796 (67.62) | 381 (32.38) |
| **Type of cooking fuel** | 0.0125 | | 0.029 | | 0.0056 | |
| safe | 4 (100.00) | 0 (0.00) | 21 (88.17) | 3 (11.83) | 25 (88.63) | 3 (11.37) |
| unsafe | 79 (50.34) | 78 (49.66) | 1160 (69.85) | 501 (30.15) | 1239 (68.18) | 578 (31.82) |

All tests used the Rao – Scott Chi-square test, ANC: Antenatal Care, NA: Not Applicable.

and Central provinces. In contrast, Southern and Luapula provinces reported the lowest rates of neonatal mortality among HIV-infected mothers (Fig 3).

The map was generated using QGIS version 3.34.1. (Source: author generated). Source of the shapefile: https://data.humdata.org/dataset/cod-ab-zmb?#. Dataset source: ZDHS women dataset accessible at: (https://www.dhsprogram.com/data).

## Spatial analysis of neonatal mortality among births to HIV-negative mothers in rural Zambia, 2018

The study revealed the geospatial patterns of neonatal mortality among births to HIV-negative mothers in rural Zambia. Specifically, Muchinga provinces emerged as a frontrunner in terms of neonatal mortality rates, with Western, Copperbelt and Lusaka provinces following closely behind. Conversely, North-western, Luapula and Central provinces exhibited the lowest rates of neonatal mortality as shown in Fig 4.

The map was generated using QGIS version 3.34.1. (Source: author generated). Source of the shapefile: https://data.humdata.org/dataset/cod-ab-zmb?#. Dataset source: ZDHS women dataset accessible at: (https://www.dhsprogram.com/data).

## A spatial autocorrelation analysis of neonatal mortality by maternal HIV status in rural Zambia using Moran's I

The spatial autocorrelation analysis using Moran's I revealed differing spatial patterns of neonatal mortality between HIV-infected and non-HIV-infected mothers in Zambia. Among HIV-infected mothers, the Moran's I statistic was −0.053

**Global Public Health**

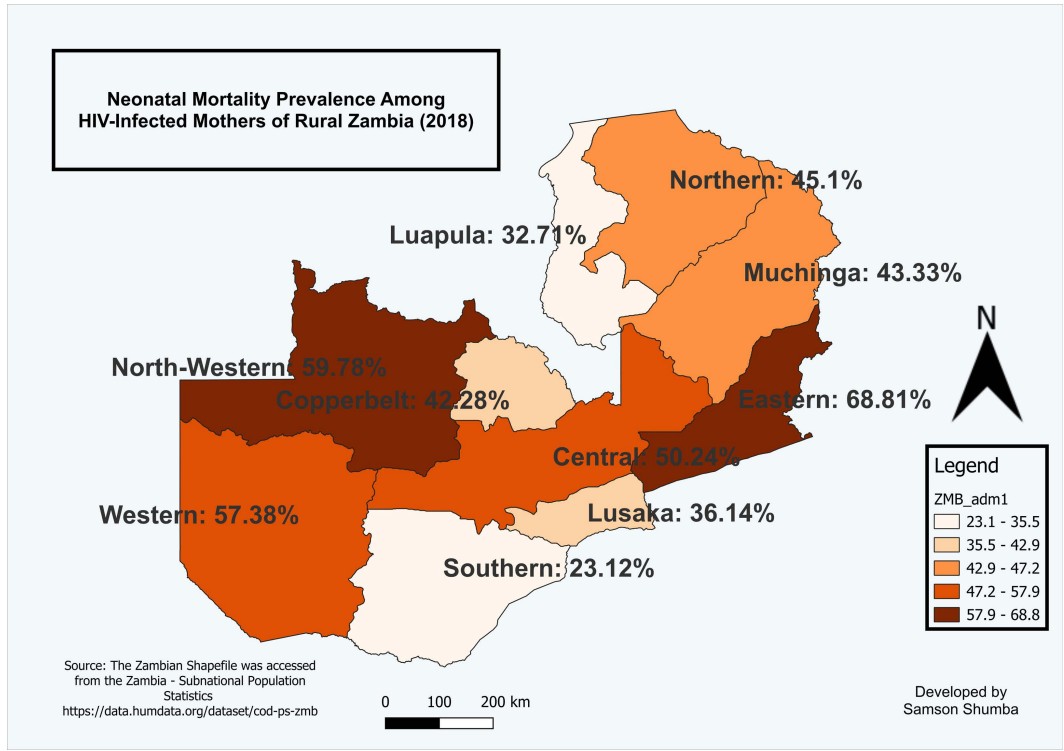

**Fig 3. Proportion of neonatal mortality among births to HIV-infected mothers in rural Zambia, 2018.**

with a p-value of $0.3904$, which is statistically insignificant. This result, along with an expected value of -0.111 and variance of 0.043, suggests a near-random spatial distribution of neonatal mortality across provinces. For instance, high neonatal mortality rates in North-Western (45.8%) and Eastern (40.3%) provinces are surrounded by provinces with varying rates such as Luapula (38.6%), Central (29.7%), and Muchinga (21.4%), indicating no clear spatial clustering or dispersion pattern.

Conversely, among non-HIV-infected mothers, the Moran's I statistic was -0.368 with a p-value of 0.8922, also statistically insignificant but suggesting a stronger tendency toward negative spatial autocorrelation compared to HIV-infected mothers. This implies that provinces with high neonatal mortality rates are often neighbored by provinces with low rates, and vice versa. For example, Muchinga Province (37.5%) is bordered by Northern (24.5%) and Central (29.7%), which have lower rates, while Eastern Province (37.2%) is adjacent to Central and Muchinga. Although the expected value remains -0.111 and variance 0.043, the lack of statistical significance in both cases underscores the need for further investigation, yet the observed spatial pattern in non-HIV-infected mothers may still reflect underlying disparities in healthcare access, quality, and socio-demographic conditions across provinces (Table 2).

## Multilevel mixed effect logistic regression of neonatal mortality among births to HIV-infected and non-infected mothers in rural Zambia

The adjusted model, reveals several significant factors associated with neonatal mortality, controlling for other variables. Among the maternal age groups, children born to mothers aged 20–24 years had increased odds of neonatal death compared to those born to mothers aged 15–19 years, with an adjusted odds ratio (AOR) of 6.11 (95% CI: 1.26–29.74).

**Fig 4. Shows the proportion of neonatal mortality among births to non HIV-infected mothers in rural Zambia (2018).**

Although other age groups did not reach statistical significance, the direction of the association varied, with older mothers such as those aged 35–39 showing a tendency toward reduced odds, holding all factors constant, albeit with wide confidence intervals.

Delivering in public health facilities was associated with a significantly increased odds of neonatal mortality compared to home deliveries, controlling for other factors (AOR = 11.34; 95% CI: 1.99–64.62). On the other hand education was strongly protective; mothers with primary education had significantly lower odds of experiencing neonatal death (AOR = 0.12; 95% CI: 0.02–0.90), and the association was even more pronounced among those with secondary or tertiary education (AOR = 0.09; 95% CI: 0.01–0.92). Birth order similarly showed no significant effects. In contrast, birthweight had a strong association with neonatal outcomes; neonates weighing 4000g or more had significantly higher odds of dying (AOR = 41.14; 95% CI: 5.92–285.94) compared to those weighing less than 2500g, holding all factors constant.

**Table 2. Spatial autocorrelation results for neonatal percentages across provinces.**

| Statistic | Neonatal mortality HIV infected mothers | Neonatal mortality non-HIV infected mothers |
|---|---|---|
| Moran's I | -0.053 | -0.368 |
| Expected value | -0.111 | -0.111 |
| Variance | 0.043 | 0.043 |
| P-value | 0.3904 | 0.8922 |

Maternal HIV status was not associated with neonatal mortality, with an AOR of 5.89 (95% CI: 0.91–38.30). Infants who were never breastfed had substantially higher odds of neonatal death (AOR = 6.85; 95% CI: 1.94–24.22), holding all other factors constant. Regarding regional variations, children born in North Western Province had significantly lower odds of neonatal death compared to those from Central Province (AOR = 0.08; 95% CI: 0.01–0.71). Other provinces showed mixed associations, with Lusaka showing a potential protective effect that was not statistically significant. Community-level factors such as community poverty, education, and employment levels were not significantly associated with neonatal mortality in this model.

In contrast, several factors were not significantly associated with neonatal mortality in this model. These included wealth status, child's sex, marital and employment status, birth order, maternal HIV status, access to improved water and sanitation, and community-level indicators such as poverty, education, and employment. While some of these variables showed directional trends (e.g., female children and mothers from richer households having lower or higher odds), the wide confidence intervals suggest substantial uncertainty around these estimates.

The intra-class correlation coefficient (ICC) for the random effect of cluster (enumeration area) was $4.31 \times 10^{-34}$, indicating that almost all variability in neonatal mortality occurred at the individual level rather than between clusters. Finally, multicolinearity diagnostics confirmed that all variables in the model had Variance Inflation Factors (VIFs) below 5, affirming the stability and reliability of the model estimates Table 3.

## Discussion

This study investigated the spatial variations and predictors of neonatal mortality among births to HIV-infected and non-infected mothers in rural Zambia, utilizing the 2018 Zambia Demographic Health Survey (ZDHS) dataset. Our analysis revealed distinctive patterns in neonatal mortality across various demographic factors shedding light on the intricate interplay of these factors influencing neonatal mortality survival and highlighting regional disparities in neonatal mortality rates in rural Zambia among HIV infected and non-infected mothers. Notably, neonatal mortality exhibited a significant association in the baseline characteristics with an increase in maternal age, reaching its peak in the 45–49 age group, accounting for 100% among HIV infected mothers compared to 95.21% in non- HIV-infected mothers.

However, utilizing multilevel logistic regression the study revealed that women aged 20–24 years had significantly increased odds of experiencing neonatal mortality compared to those aged 15–19 years, after adjusting for other variables. This finding contrasts with much of the existing literature, which often reports higher neonatal mortality rates among adolescent mothers due to biological immaturity, limited access to maternal healthcare, and socio-economic disadvantages [34,35]. However, some studies suggest that first-time births at older adolescent ages may involve more caution, closer monitoring, or higher utilization of antenatal services, potentially reducing the risk of neonatal death [36]. The elevated risk among mothers aged 20–24 in this study could be attributed to contextual factors such as increased parity, reduced social support, or differences in health-seeking behavior not captured in the model. It also raises the possibility that adolescent mothers may be receiving more targeted maternal and child health interventions due to their classification as a high-risk group, which may inadvertently lead to better neonatal outcomes in this demographic [37].

Maternal education emerged as a significant protective factor against neonatal mortality in this study. Mothers with primary education had markedly reduced odds of experiencing neonatal death (AOR = 0.12; 95% CI: 0.02–0.90), and this protective association was even stronger among those with secondary or tertiary education (AOR = 0.09; 95% CI: 0.01–0.92). The observed relationship between maternal education and neonatal mortality aligns with global evidence. A systematic review and meta-analysis published in *The Lancet* revealed a clear dose–response relationship, where completion of 12 years of maternal education (equivalent to secondary school) was associated with a 31% reduction in under-five mortality compared to mothers with no formal education. Furthermore, each additional year of maternal schooling was linked to an approximate 3% decline in child mortality. Notably, the impact of maternal education on child survival was found to be stronger than that of paternal education, suggesting profound transgenerational benefits for child health [38].

**Table 3. Multilevel mixed effect logistic regression of neonatal mortality among births to HIV-infected and non-infected mothers in rural Zambia.**

| Variables | Model 1 | Model 2 | Model 3 |
|---|---|---|---|
| | OR | AOR | AOR |
| **Intercept** | | | |
| Intercept 1 | 0.83 | 0.33 (0.02 - 4.85) | 0.13 (0.004 - 4.83) |
| Intercept 2 | 1.08 | -- | -- |
| **Mother's age** | | | |
| 15 - 19 | | Ref (1) | Ref (1) |
| 20 - 24 | | 3.11 (0.58 - 16.70) | 6.11 (1.26 - 29.74)** |
| 25 -29 | | 1.28 (0.22 - 7.35) | 3.22 (0.48 - 21.78) |
| 30 - 34 | | 1.67 (0.21 - 13.33) | 2.49 (0.32 - 19.43) |
| 35 - 39 | | 0.12 (0.02 - 0.91)* | 0.14 (0.01 - 1.62) |
| 40 -44 | | 1.94 (0.09 - 41.88) | 11.52 (0.53 - 248.95) |
| 45 - 49 | | -- | -- |
| **Wealth Index** | | | |
| Poorest | | Ref (1) | Ref (1) |
| Poorer | | 2.57 (0.81 - 8.13) | 3.46 (0.75 - 16.00) |
| Middle | | 2.68 (0.81 - 8.91) | 2.87 (0.56 - 14.79) |
| Richer | | 2.30 (0.28 - 18.74) | 9.87 (0.58 - 167.90) |
| Richest | | 0.47 (0.02 - 13.73) | 6.37 (0.15 - 278.10) |
| **Sex of child** | | | |
| Male | | Ref (1) | Ref (1) |
| Female | | 0.25 (0.10 - 0.62)*** | 0.44 (0.17 - 1.13) |
| **Place of Delivery** | | | |
| Home | | Ref (1) | Ref (1) |
| Private | | 0.16 (0.01 - 4.03) | 0.23 (0.01 - 10.06) |
| Public | | 7.08 (1.51 - 33.11)** | 11.34 (1.99 - 64.62)*** |
| **Education** | | | |
| None | | Ref (1) | Ref (1) |
| Primary | | 0.22 (0.04 - 1.23) | 0.12 (0.02 - 0.90)* |
| Secondary/tertiary | | 0.28 (0.04 - 1.94) | 0.09 (0.01 - 0.92)* |
| **Marital status** | | | |
| Not married | | Ref (1) | Ref (1) |
| Married | | 2.56 (0.97 - 6.80) | 2.22 (0.67 - 7.35) |
| **Employment** | | | |
| No | | Ref (1) | Ref (1) |
| Yes | | 1.18 (0.54 - 2.56) | 1.42 (0.46 - 4.38) |
| **Birth order** | | | |
| 0 - 3 | | Ref (1) | Ref (1) |
| 4 - 6 | | 0.42 (0.09 - 1.89) | 0.28 (0.05 - 1.63) |
| 7 + | | 4.06 (0.73 - 22.56) | 2.69 (0.34 - 21.21) |
| **Birthweight** | | | |
| <2500g | | Ref (1) | Ref (1) |
| 2500g - 3999g | | 0.57 (0.18 - 1.79) | 0.98 (0.26 - 3.68) |
| 4000g and above | | 14.19 (2.25 - 89.39) | 41.14 (5.92 - 285.94)*** |

*(Continued)*

**Table 3.** (Continued)

| Variables | Model 1 | Model 2 | Model 3 |
|---|---|---|---|
| | OR | AOR | AOR |
| **HIV status** | | | |
| Negative | | Ref (1) | Ref (1) |
| Positive | | 1.86 (0.46 - 7.44) | 5.89 (0.91 - 38.30) |
| **Toilet** | | | |
| Improved | | Ref (1) | Ref (1) |
| Unimproved | | 0.92 (0.37 - 2.25) | 1.61 (0.46 - 5.67) |
| **Duration of breastfeeding** | | | |
| Ever breastfed, | | Ref (1) | Ref (1) |
| Never breastfed | | 3.52 (1.32 - 9.39) | 6.85 (1.94 - 24.22)*** |
| **Source of drinking water** | | | |
| Improved | | Ref (1) | Ref (1) |
| Unimproved | | 0.85 (0.36 - 2.00) | 1.29 (0.45 - 3.70) |
| **Contextual Factors** | | | |
| **Region** | | | |
| Central | | | Ref (1) |
| Copperbelt | | | 3.45 (0.14 - 86.02) |
| Eastern | | | 3.36 (0.32 - 35.15) |
| Luapula | | | 3.92 (0.41 - 37.87) |
| Lusaka | | | 0.05 (0.002 - 1.46) |
| Muchinga | | | 3.53 (0.48 - 26.03) |
| Northern | | | 2.11 (0.23 - 19.86) |
| North Western | | | 0.08 (0.01 - 0.71) |
| Southern | | | 1.40 (0.19 - 10.49) |
| Western | | | 0.29 (0.04 - 1.85) |
| **Community Poverty** | | | |
| Low | | | Ref (1) |
| Middle | | | 0.59 (0.11 - 3.22) |
| High | | | 1.21 (0.23 - 6.33) |
| **Community Education** | | | |
| Low | | | Ref (1) |
| Middle | | | 0.35 (0.31 - 5.89) |
| High | | | 0.24 (0.4 - 1.32) |
| **Community Employment status** | | | |
| Low | | | Ref (1) |
| Middle | | | 0.23 (0.05 - 1.08) |
| High | | | 1.03 (0.29 - 3.64) |
| **Diagnostic tests** | | | |
| AIC | 239.208 | 235.606 | 235.2171 |
| BIC | 245.503 | 325.44 | 373.4299 |
| VIF | -- | <5 | <5 |

***= P-value ≤ 0.01

**= 0.01 <P-value ≤ 0.03

*=0.03<P-value<0.05 AOR: Adjusted Odds Ratio

These findings support global recommendations emphasizing the importance of education as a social determinant of health. The World Health Organization's 2023 Progress Report on Maternal and Newborn Health advocates for targeted investments in female education as part of a holistic strategy to reduce neonatal mortality. The report highlights that improving maternal education—alongside strengthening healthcare systems—can accelerate progress toward achieving global maternal and child health goals [39].

Unexpectedly, delivering in public health facilities was associated with significantly higher odds of neonatal mortality compared to home deliveries, after adjusting for other factors. This counterintuitive finding has been observed in other low-resource settings and may reflect systemic challenges within public healthcare systems rather than the intrinsic risk associated with facility-based delivery. These findings are in line with research from Nigeria by Adedini et al. (2014), which highlighted that delivering in a health facility does not automatically translate into better neonatal outcomes, particularly where facilities are under-resourced and staff are inadequately trained [40]. Similarly, a recent study in Zambia by Mumba et al. (2025) found that poor service quality in public health facilities, where most women deliver, continues to undermine maternal and neonatal health outcomes despite increased facility-based deliveries [41].

One plausible explanation is that women who deliver in public health facilities often do so as a last resort when complications have already developed, thereby increasing the risk of adverse neonatal outcomes. This scenario reflects a pattern of delayed care-seeking, where facilities are accessed too late to prevent fatal outcomes. Moreover, systemic weaknesses such as staff shortages, inadequate neonatal resuscitation skills, lack of essential drugs, and suboptimal infection control practices may further compromise newborn survival in such settings. This finding underscores the need to move beyond promoting facility deliveries as a stand-alone intervention and instead focus on improving the quality of care within health institutions. The WHO's "Every Newborn Action Plan" emphasizes that the quality of care not just access is critical to reducing neonatal deaths. Investments in health worker training, infrastructure, accountability mechanisms, and respectful maternity care are essential to ensure that facility-based deliveries translate into improved neonatal survival [42].

This study also explored the relationship between maternal HIV status and neonatal mortality in Zambia using an adjusted multivariable model to control for potential confounders. The analysis revealed that maternal HIV status was not significantly associated with neonatal mortality. Although this finding may seem surprising given the biological vulnerability of neonates born to HIV-positive mothers, it is consistent with a growing body of literature that highlights the nuanced and context-dependent nature of this relationship. Several studies have documented an elevated risk of neonatal and infant mortality among children born to HIV-positive mothers, largely due to factors such as maternal immunosuppression, opportunistic infections, and increased likelihood of preterm birth or low birth weight [43]. However, the introduction and scale-up of antiretroviral therapy (ART) programs have significantly altered this landscape. ART not only reduces maternal viral load—thereby lowering the risk of vertical transmission but also improves maternal health, pregnancy outcomes, and neonatal survival [44].

In contexts like Zambia, where prevention of mother-to-child transmission (PMTCT) programs have been widely implemented, the impact of maternal HIV on neonatal mortality may be attenuated. Studies have shown that when HIV-positive mothers receive timely and sustained ART, neonatal outcomes are comparable to those of HIV-negative mothers [45]. Moreover, increased access to skilled birth attendance, early infant diagnosis, and integration of HIV care with maternal and child health services have collectively contributed to improved neonatal health outcomes in HIV-exposed infants. Therefore, the lack of a statistically significant association in this study may reflect the effectiveness of Zambia's national PMTCT initiatives and ART coverage. These findings emphasizes the importance of maintaining and strengthening HIV care and treatment services as an integral component of maternal and neonatal health strategies. Continued investment in comprehensive HIV care, including adherence support and health system strengthening, remains vital to sustaining the progress made in reducing neonatal mortality among HIV-exposed infants.

The spatial analysis of neonatal mortality in Zambia revealed distinct patterns for HIV-infected and non-HIV-infected mothers. For HIV-infected mothers, the Moran's I statistic was close to zero, indicating a nearly random distribution of neonatal mortality across provinces. This suggests that high neonatal mortality rates in some provinces are surrounded by those with varying rates, with no clear spatial clustering. Similarly, the Moran's I statistic for non-HIV-infected mothers also showed negative spatial autocorrelation, suggesting that provinces with high neonatal mortality rates are often neighbored by those with lower rates, and vice versa. These results align with findings from studies in other settings, such as Getis (2007) [46], who reported similar negative spatial autocorrelation in areas with varying resource allocation and service delivery.

These patterns of negative spatial autocorrelation in both groups, though statistically insignificant, could reflect underlying disparities in healthcare access, infrastructure, and socio-economic conditions between provinces. Such findings point to a fragmented or competitive spatial structure, where regions with varying resource allocations and demographic factors are not clustering together. This underscores the need for further investigation into the structural and service delivery disparities that may influence neonatal outcomes across Zambia's diverse provinces [47].

Furthermore our study emphasizes the multifaceted nature of factors influencing neonatal mortality, including maternal age, neonate sex, maternal HIV status, and duration of breastfeeding, place of delivery and Region (North Western province). This intricate web of variables highlights the need for a shift in focus within current mother and child health programs to incorporate activities that empower women to make informed decisions regarding the timing of childbirth, monitor their HIV status, duration of breastfeeding and consider the birthweight and sex of their child. Additionally, a systematic review and meta-analysis conducted in sub-Saharan Africa revealed that rural residence significantly contributes to high neonatal mortality rates [48]. Our study's novel findings offer valuable insights previously unexplored, illuminating the intricate landscape of neonatal mortality in rural Zambia. Recognizing the distinct patterns of neonatal mortality between HIV-infected and non-infected mothers is crucial for implementing tailored interventions that enhance maternal and child health outcomes within these communities.

## Study strength and study limitations

This study benefits from the utilization of national data, offering a representative sample of neonates, thus enabling generalizability to this specific demographic. However, certain limitations must be acknowledged. The study relies on the 2018 Zambia Demographic and Health Survey (ZDHS) dataset, employing a cross-sectional design, which indicates correlation rather than causation between the outcome and individual or contextual factors. Additionally, the utilization of contextual and community-level factors from the ZDHS may not fully capture the community experience. Moreover, potential recall bias exists, with mothers possibly misreporting crucial information like their child's age at death, birth order, and birth weight. To address this, rigorous data cleaning procedures were implemented to identify and rectify inconsistencies or outliers.

## Policy recommendation

This study emphasizes the urgent need for targeted health policy reforms to address the complex and region-specific factors influencing neonatal mortality in rural Zambia. While adolescent pregnancies are traditionally viewed as high-risk, findings revealed that women aged 20–24 years had significantly higher odds of neonatal mortality, suggesting the need to broaden maternal health interventions to include this age group. Additionally, maternal education proved to be a strong protective factor, highlighting the importance of reinforcing policies that promote girls' education and integrate health education into school curricula. Unexpectedly, public health facilities were associated with higher neonatal mortality, pointing to critical gaps in the quality of care. This calls for urgent investments in staff training, infrastructure, and accountability within public health institutions to ensure that facility-based deliveries result in improved outcomes.

Moreover, while maternal HIV status was not significantly associated with neonatal mortality, this may reflect the success of Zambia's PMTCT and ART programs. Sustaining and expanding these programs is essential, especially in underserved rural areas. Spatial analysis revealed regional disparities, emphasizing the need for data-driven, location-specific interventions and equitable distribution of health resources. Strengthening community-based health education and empowering women with knowledge on safe childbirth, breastfeeding, and neonatal care are vital. These policy actions align with Zambia's National Health Strategic Plan and the RMNCAH&N framework and are crucial to reducing neonatal deaths and advancing maternal and child health equity in rural Zambia.

## Conclusion

In conclusion, this study emphasizes significant geospatial variations in neonatal mortality rates among HIV-infected and non-infected mothers across rural Zambia, emphasizing the need for targeted interventions to address these disparities. The findings reveal that regional differences in access to healthcare, socio-economic conditions, maternal education levels, and cultural practices significantly influence neonatal outcomes. Addressing these multifaceted factors is crucial for improving maternal and child health in Zambia. By enhancing healthcare access, increasing the availability of antiretroviral therapy, and implementing culturally sensitive educational campaigns, stakeholders can work collaboratively to reduce neonatal mortality rates. Ultimately, a comprehensive approach that considers the unique challenges faced by different provinces will be essential for achieving equitable health outcomes and promoting the well-being of mothers and their infants throughout the country.

## Acknowledgments

Many thanks to the Zambia Statistical Agency (ZSA) and the DHS program for granting permission to utilize the 2018 ZDHS. **Dedication:** This paper is dedicated, with love and reverence, to the cherished memory of Wise Mwila Katumbo.

## Author contributions

**Conceptualization:** Samson Shumba.

**Data curation:** Samson Shumba.

**Formal analysis:** Samson Shumba.

**Methodology:** Samson Shumba, Deborah Tembo, Miyanda Simwaka, Nedah Chikonde Musonda, Sebean Mayimbo.

**Software:** Samson Shumba.

**Supervision:** Sebean Mayimbo.

**Visualization:** Samson Shumba.

**Writing – original draft:** Samson Shumba, Deborah Tembo, Miyanda Simwaka, Nedah Chikonde Musonda, Chipo Nkwemu, Sebean Mayimbo.

**Writing – review & editing:** Samson Shumba, Deborah Tembo, Miyanda Simwaka, Nedah Chikonde Musonda, Chipo Nkwemu, Sebean Mayimbo.

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
