## [Decision Letter · Decision Letter 0]

Geospatial patterns and predictors of neonatal mortality among HIV infected and non-infected mothers of rural Zambia: A comparative analysis of the 2018 Zambia Demographic and Health Survey

Dear Dr. Shumba,

The reviewers has provided their comments below. They have requested better clarity around the methodology used and statistical analyses. Please review their comments and make the appropriate revisions.

Kind regards,

Emma Campbell, Ph.D

Staff Editor

Reviewers' comments:

**Comments to the Author**

1. Does this manuscript meet PLOS Global Public Health’s publication criteriaReviewer #1: No?

 2. Has the statistical analysis been performed appropriately and rigorously? Reviewer #1: No?

 3. Have the authors made all data underlying the findings in their manuscript fully available (please refer to the Data Availability Statement at the start of the manuscript PDF file)?

 4. Is the manuscript presented in an intelligible fashion and written in standard English?

Title: Too crowded and not clear enough.

Abstract:

Needs to be more concise and is poorly written. For example, the first two sentences in the background below don’t belong in abstract.

“Neonatal mortality is the death of a live-born infant within the first 28 completed days of life. Neonatal mortality remains a major public health concern in most African countries, with the Sub-Saharan region being the highest contributor at 27 deaths per 1000 live births, accounting for 43% of the total number of deaths.”

Methods

More specifically the methods section lacks clarity on why spefic variables were chosen, conceptually how they affect the outcomes, the rationale for the methods including choice of mixed effects model.

On Community level variables; Community-level variables in this study were derived by aggregating individual-level data into clusters and incorporated community poverty, community education, community knowledge of family planning (FP) methods, and place of residence.

Any reason why this was done and why important it is to generalise assessment of range/ICC?

Data Analysis

For descriptive purposes, frequencies and percentages were computed for categorical variables. To determine the association between the outcome variable (neonatal mortality) and the categorical variables, the Uncorrelated Design Based Chi-square test (Rao – Scott Chi-square test) was used. The multilevel mixed-effect logistic regression model was employed to determine the factors affecting newborn mortality. Additionally, the study used an investigator-led approach, all variables were selected from a wide range of literature. The Intra-Class Correlation (ICC), Akaike Information Criteria (AIC), and the Bayesian Information Criteria (BIC) were sufficiently explored to select the best-fit model. Stata version 14.2 was used for analysis.

Not clear what analysis was done. The authors alude they did AIC and BIC to perform model selection; but was the model/variable (at this point it is very confusing as to what the paper is all about) selection fit into a final model? And why was it done?

In description of Multilevel logistic regression, was it a multivariable model? what were the levels in the hierarchy and what were the fixed effects, what were the random effects?

In the description of the Multilevel Logistic Regression model, the following points need to be addressed for clarity:

Multivariable Model: if the model included multiple predictors.

- Levels in the Hierarchy: Clearly define the hierarchical structure:

- Level 1: Individual level (e.g., neonates).

- Level 2: Cluster level (e.g., health facilities or geographical regions).

Fixed Effects: Specify the predictors considered as fixed effects, such as:

- Socio-demographic factors (e.g., maternal age, education level, household income).

- Health-related variables (e.g., prenatal care visits, birth weight, delivery method).

Random Effects: Identify the random effects, which account for variability at the cluster level:

- Random intercepts (e.g., differences in neonatal mortality rates across clusters).

- Random slopes for some predictors, if applicable (e.g., varying effects of prenatal care visits across different regions).

Providing this detailed information will enhance the clarity and comprehensiveness of the methods section.

What was the outcome and how were the variables in the final model(s) chosen?

Did the authors assess any correlation in variables? Some variables could have multicollinearity

Geospatial Analysis

To examine the clustering and spatial distribution of neonatal mortality across provinces in Zambia, this study utilized Quantum Geographic Information System (QGIS) version 3.34.1 to construct a univariate choropleth map. Spatial analysis was conducted at the provincial level, with each neonate matched to their provincial residence using geo-coordinate data collected from the Demographic and Health Survey (DHS). The DHS employs pre-defined data assigning each case to a province. Spatial analysis units were defined as clusters of sample households as designated by the Zambian Demographic and Health Survey (ZDHS). The coordinate system utilized was the World Geographic System (WGS) 1984 Universal Transverse Mercator (UTM) Zone 36S.

Not much analysis here; I recommend saying mapping of neonatal mortality was done in QGIS….

Results.

Furthermore, mortality rates were significantly elevated among non-HIV-infected mothers lacking formal education (40.78%), compared to 54.56% among HIV-infected mothers

Statement doesn’t make sense at all

Overall, neonatal mortality stood at 41.13% among mothers (both HIV and non-HIV-infected) with no formal education.

This is a summary and should come in a table or initially when giving results

The neonatal mortality rate was high at 37.17% among neonates whose mothers (both HIV infected and non-infected mothers) were reported to be working.

What makes 37.17% high in comparison to what?

The so called geospatial analysis can be a table. Also, whats the rationalled for the classes in the map? And what makes >57.87 high?

**Do you want your identity to be public for this peer review?** For information about this choice, including consent withdrawal, please see our Privacy Policy

---

## [Decision Letter · Decision Letter 1]

PGPH-D-24-01165R1

Spatial Variations and Predictors of Neonatal Mortality among HIV-Infected and Non-Infected Mothers in Rural Zambia

Dear Dr. Shumba,

Thank you for submitting your manuscript to PLOS Global Public Health. After careful consideration, we feel that it has merit but does not fully meet PLOS Global Public Health’s publication criteria as it currently stands. Therefore, we invite you to submit a revised version of the manuscript that addresses the points raised during the review process.

The use of "geospatial variations" is misleading, as the methods section does not describe an appropriate analytical approach to support this claim. The maps presented do not effectively communicate a clear message beyond displaying different color codes, and the statistical significance of the observed patterns is unclear. The authors should consider using Moran’s I to identify spatial clusters of areas with similar risks for the outcome of interest.

Regarding the multilevel modeling approach, it is standard practice to run three models. The first is a **null model** , which includes only the response variable and is used to estimate variance. **Model 1** is a fixed-effects model that controls for individual-level factors while incorporating random effects. **Model 2** includes contextual factors along with the random effects, and the **final model** integrates all individual and contextual factors while retaining the random effects. The current analysis does not follow this structured approach, and the methodology should be revised accordingly.

Additionally, the interpretation of odds ratios is incorrect and requires a clearer and more structured explanation. Finally, the manuscript contains several grammatical errors, and a thorough review and revision for language clarity would be helpful.

We look forward to receiving your revised manuscript.

Kind regards,

Julius Odhiambo

Academic Editor

Journal Requirements:

Additional Editor Comments (if provided):

Reviewers' comments:

Reviewer's Responses to Questions

**Comments to the Author**

Reviewer #2: All comments have been addressed

Reviewer #3: (No Response)

publication criteria?

Reviewer #2: Yes

Reviewer #3: Yes

3. Has the statistical analysis been performed appropriately and rigorously?

Reviewer #2: Yes

Reviewer #3: No

4. Have the authors made all data underlying the findings in their manuscript fully available (please refer to the Data Availability Statement at the start of the manuscript PDF file)?

Reviewer #2: No

Reviewer #3: Yes

5. Is the manuscript presented in an intelligible fashion and written in standard English?

Reviewer #2: No

Reviewer #3: No

Reviewer #2: Dear Author,

The manuscript is well written.

Reviewer #3: While the paper presents an interesting topic, it will benefit from an in-depth language and grammatical editing from colleagues from school of Education (University of Zambia). Further, the science needs to be recast. The model, hierarchical or multilevel model has to be re-run and i have provided two useful resources that can be used.

**Do you want your identity to be public for this peer review?** For information about this choice, including consent withdrawal, please see our Privacy Policy

Reviewer #2: **Yes: ** Prof. Dr Sunil Raja Manandhar

Reviewer #3: **Yes: ** Chester Kalinda, PhD

---

## [Editor Report · Decision Letter 2]

PGPH-D-24-01165R2

Spatial Variations and Predictors of Neonatal Mortality among HIV-Infected and Non-Infected Mothers in Rural Zambia

Dear Dr. Shumba,

Thank you for submitting your manuscript to PLOS Global Public Health. After careful consideration, we feel that it has merit but does not fully meet PLOS Global Public Health’s publication criteria as it currently stands. Therefore, we invite you to submit a revised version of the manuscript that addresses the points raised during the review process.

We look forward to receiving your revised manuscript.

Kind regards,

Julius Odhiambo

Academic Editor

Journal Requirements:

Additional Editor Comments (if provided):

The stated study objective, "geospatial patterns and predictors of neonatal mortality among both HIV-infected and non-infected mothers in rural Zambia," suggests that the unit of analysis is the mother. However, the analysis appears to focus on neonatal outcomes. I recommend rewording the objective for clarity and consistency with the analysis — for example:

"To examine the geospatial patterns and predictors of neonatal mortality among births to HIV-infected and non-infected mothers in rural Zambia."

This refinement more accurately reflects that the outcome is at the neonatal level rather than the maternal level.

I recommend clarifying early in the manuscript whether the primary geospatial unit of analysis is the cluster, subnational area, or both. If both levels are important, please specify this explicitly and justify the approach, since patterns at the cluster level may differ substantially from aggregated subnational patterns (This might as well inform the approach).
---

## [Decision Letter · Decision Letter 3]

Spatial Variations and Predictors of Neonatal Mortality among Births to HIV-Infected and Non-Infected Mothers in Rural Zambia

PGPH-D-24-01165R3

Dear Mr. Shumba,

We are pleased to inform you that your manuscript 'Spatial Variations and Predictors of Neonatal Mortality among Births to HIV-Infected and Non-Infected Mothers in Rural Zambia' has been provisionally accepted for publication in PLOS Global Public Health.

Best regards,

Julius Odhiambo

Academic Editor

Reviewer Comments (if any, and for reference):

Reviewer's Responses to Questions

**Comments to the Author**

Reviewer #3: All comments have been addressed

publication criteria?

Reviewer #3: Yes

3. Has the statistical analysis been performed appropriately and rigorously?

Reviewer #3: Yes

4. Have the authors made all data underlying the findings in their manuscript fully available (please refer to the Data Availability Statement at the start of the manuscript PDF file)?

Reviewer #3: (No Response)

5. Is the manuscript presented in an intelligible fashion and written in standard English?

Reviewer #3: Yes

Reviewer #3: The manuscript that was submitted had no highlights to show where changes had been made.

**Do you want your identity to be public for this peer review?** For information about this choice, including consent withdrawal, please see our Privacy Policy

Reviewer #3: **Yes: ** Chester Kalinda (PhD)
